# Distinct Amphibian Elevational and Seasonal Phylogenetic Structures Are Determined by Microhabitat Variables in Temperate Montane Streams

**DOI:** 10.3390/ani12131673

**Published:** 2022-06-29

**Authors:** Xi-Wen Peng, Jing Lan, Zi-Jian Sun, Wen-Bo Zhu, Tian Zhao

**Affiliations:** 1College of Fisheries, Southwest University, Chongqing 400715, China; pxw@swu.edu.cn (X.-W.P.); lan97hhh@outlook.com (J.L.); sunzj19@outlook.com (Z.-J.S.); 2CAS Key Laboratory of Mountain Ecological Restoration and Bioresource Utilization & Ecological Restoration Biodiversity Conservation Key Laboratory of Sichuan Province, Chengdu Institute of Biology, Chinese Academy of Sciences, Chengdu 610041, China; wenbo_zhu2022@163.com; 3Central South Inventory and Planning Institute of National Forestry and Grassland Administration, Changsha 410014, China

**Keywords:** phylogenetic diversity, amphibian conservation, community assembly, elevational patterns, seasonal change

## Abstract

**Simple Summary:**

Phylogenetic structure can be used to understand various ecological patterns. The main objective of the present study was to understand the elevational and seasonal amphibian phylogenetic structures in temperate montane streams. This study was conducted in 13 streams located in lowland and highland sites of Tianping mountain, China, in April, June, August, and October 2017, separately. We found that the elevational spatial patterns were not significantly different, but the seasonal temporal patterns differed significantly for amphibian phylogenetic structures, associated with the variation of microhabitat variables.

**Abstract:**

Phylogenetic structure is a key facet of biodiversity, reflecting the evolutionary history of species, and thus can be used to understand various ecological patterns. Although amphibian phylogenetic structures have been tested across space and time separately, simultaneous quantifications are still needed. In the present study, amphibians in streams of Tianping mountain, China, were selected as the model to investigate their elevational spatial and seasonal temporal patterns of phylogenetic diversity. Specifically, 13 streams located in lowland and highland sites were sampled for amphibians and measured for microhabitat variables in April, June, August, and October 2017, separately. Four phylogenetic structural indices, including Faith’s PD, standardized effect size (SES) of Faith’s PD, mean pairwise phylogenetic distance index (MPD), and SES.MPD, were calculated. Our results revealed that amphibian phylogenetic patterns were not significantly different between lowland and highland sites, but differed significantly between four seasons, associated with distinct community assembly rules (phylogenetically overdispersed vs. phylogenetically clustered). Importantly, these patterns were strongly determined by microhabitat variables such as rock cover, water temperature, and water depth. Our results provide fundamental knowledge to better protect amphibian diversity. Both elevational and seasonal variations are important to understanding the general patterns of amphibian community assembly rules.

## 1. Introduction

Biological diversity is increasingly recognized to be important for human society. This is because it has strong positive relationships with ecosystem functioning, with higher diversity promoting the efficiency of a community in capturing resources [1]. Traditionally, biodiversity is assessed based on the species taxonomic facet, which is considered to be a fundamental discipline [2]. However, based on the claims that it is the functional traits and evolutionary processes driving species performances in ecosystems, increasing studies have started to quantify other important facets of biodiversity (e.g., functional and phylogenetic structures) [3]. This is especially true for phylogenetic diversity, which reflects the evolutionary history of species and thus can be used to understand various ecological patterns [4]. Specifically, phylogenetic structures have been widely used in two main aspects during past decades, including identifying the priority areas for biodiversity conservation [5,6,7,8] and understanding the mechanisms driving the co-occurrence of species [9,10,11].

Typically, biodiversity is quantified across space and time. This is because space and time are the two most important components of the ecological niche, which can help ecologists to better understand community dynamics, as well as the habitats and resource utilization of species. Based on the scale of interest, the spatial patterns of biodiversity can be investigated between elevational sites or geographical regions. For instance, Wang et al. [12] investigated the elevational patterns of amphibian phylogenetic structures in Mount Emei, China, and they suggested that the main mechanism underlying amphibian assembly processes shifted from environmental filtering to competitive exclusion with increasing elevations. Hu et al. [13] analyzed the phylogenetic diversity of terrestrial vertebrates in China, with higher values being detected in South and Southwest China. Therefore, these areas were considered as hotspots with high conservation priority. In terms of temporal patterns, biodiversity is usually assessed between years or seasons, in particular for amphibians. This is because amphibian evolutionary dynamics, breeding, and migration can be strongly affected by the fluctuations in climate factors between years or seasons. For instance, long-term phylogenetic structures can provide a comprehensive overview of the history of amphibian diversification [14]. The seasonal fluctuation of phylogenetic diversity can show the changes in amphibian ecological properties (e.g., assembly rules) [15]. Although quantitative studies have been conducted to reveal the spatial and temporal patterns of amphibian phylogenetic diversity separately, simultaneous quantifications are still needed. This is especially true when focusing on the local scale (e.g., montane streams), in which both space and time can affect the distributions and activities of amphibian species [16].

Many factors contribute to the detection of spatial and temporal patterns of amphibian diversity. For phylogenetic diversity, species richness should be considered as the prior determinant [13]. Moreover, climatic variables are also regarded as key factors. For instance, humidity levels were positively related to amphibian phylogenetic diversity in Brazilian forests [17]. Temperature, precipitation, and ecosystem energy were also strongly associated with amphibian phylogenetic diversity in tropical regions [18]. However, most of these studies were conducted at the regional or global scales (i.e., large geographic scale). Evidence is still needed to reveal the microhabitat determinants of amphibian phylogenetic diversity at a local scale, which can help us better understand the species assemblage rules in a single place. Recently, studies have been conducted to reveal that amphibian distributions can be affected by microhabitat factors such as water temperature and leaf litter depth [16,19]. Therefore, we hypothesize that microhabitat variables can also induce cascading effects on amphibian phylogenetic diversity.

In the present study, we investigated the elevational spatial and seasonal temporal patterns of amphibian phylogenetic diversity in temperate montane streams. Specifically, we first investigated amphibian assemblages between elevational areas and between seasons, separately. We then assessed the potential difference in amphibian phylogenetic diversity between elevational sites and between seasons, separately. Finally, we revealed the microhabitat determinants of amphibian phylogenetic diversity.

## 2. Materials and Methods

### 2.1. Study Area

The present study was conducted at temperate montane streams in Tianping mountain (about 20,000 ha), northwest Hunan Province, China (29.714072°–29.787100° N, 109.906154°–110.170800° E). This area belongs to the core region of Badagongshan National Nature Reserves, with the elevation changing from 300 to 1890 m. Two climatic zones coinciding with distinct vegetation cover can be detected from low to high elevations. Specifically, it is relatively warm (mean annual temperature: 13.7–15.9 °C) in the low-elevational area (300–1000 m), with crops and evergreen broad-leaved forests dominating this area. In contrast, the mean annual temperature is below 10.0 °C in the high-elevational area (1000–1890 m), in which the main vegetation cover is evergreen deciduous broadleaf forests [20].

### 2.2. Amphibian Sampling

We randomly selected 13 montane streams as the transects, including five low-elevational transects and eight high-elevational transects (Appendix A). The length and width of the transects were 200 m × 2 m. Considering the spatial autocorrelation, these transects were separated from each other by a deep gorge or other prominent landmarks, with a minimum distance of 1.5 km. Amphibian sampling was conducted in April, June, August, and October 2017, separately, in accordance with four distinct seasons in this area (i.e., spring, early summer, midsummer, and autumn). These seasons covered the main activities of amphibians such as breeding, foraging, and migration. The combination of distance sampling and quadrat sampling approaches was used to search amphibians in the transects based on the nocturnal time-constrained visual encounter surveys. Details of the sampling protocols were provided in Zhu et al. [16] and Sun et al. [21]. All the captured individuals were identified as species based on the external morphology following Fei et al. [21,22]. We did toe clips for five individuals per species randomly, which were preserved in 95% ethanol immediately for further analyses. Finally, all the amphibians were released back to the habitat where they were captured.

### 2.3. Microhabitat Variables

A set of 15 microhabitat variables were measured in each transect during the sampling events in April, June, August, and October, separately. Specifically, these variables included air temperature (°C), which was measured by using a mercury thermometer at 2 m above the ground. The elevation (m) of each transect was recorded using a GPS (ICEGPS 660). Air humidity (%), canopy cover (%), water temperature (°C), water pH, water conductivity (S/m), and current velocity (m/s) were measured by using portable instruments. Water depth (cm), water width (m), and leaf litter depth (cm) were measured using steel tape. Number of trees, shrub cover (%), leaf litter cover (%), and rock cover (%) were recorded or estimated by the same person. Details of the measurement approaches can be found in Zhu et al. [16]. These variables were selected based on previous studies indicating that they can potentially determine the distribution of amphibians, and thus amphibian diversity [16,19,21,22].

### 2.4. Phylogenetic Tree

A phylogenetic tree was constructed based on a supermatrix generated from two sequences (i.e., 16S rRNA and Cytochrome c Oxidase Subunit I) of all the species observed in the field (Appendix A). All of these sequences were obtained from the toe tissue samples we preserved following Khatiwada et al. [23]. These sequences have been uploaded to the National Center of Biotechnology Information (NCBI: https://www.ncbi.nlm.gov (accessed on 1 June 2022)).

### 2.5. Statistical Analyses

Phylogenetic diversity was represented by four indices. Specifically, we first calculated the value of Faith’s PD, which was the sum of all branch lengths of the phylogeny tree connecting all species in each transect [5]. Since Faith’s PD can be affected by the number of species, we also calculated the standardized effect size (SES) of Faith’s PD in each transect. Specifically, all the species detected in the field were gathered as the regional species pool. Null models were run for each transect by randomly selecting species 999 times from this pool, and the number of species generated was the same as that observed in each transect [24].

We also calculated the mean pairwise phylogenetic distance (MPD) index, which reflected the average phylogenetic distance between all pairs of amphibian species detected in each transect. Moreover, the SES of MPD was calculated based on the equation:$$\mathrm{SES}.\mathrm{MPD}=-\frac{{\mathrm{meanMPD}}_{\mathrm{ob}}-{\;\mathrm{meanMPD}}_{\mathrm{ra}}}{{\mathrm{sdMPD}}_{\mathrm{ra}}}$$
where mean MPDob is the observed value of mean pairwise phylogenetic distance. The mean MPDra is the mean value from 999 randomly generated amphibian assemblages where species were randomly shuffled. The sdMPDra is the standard deviation of the null distribution. This index can be used to assess the mechanism underlying amphibian assembly processes, with a positive value indicating the phylogenetically clustered and a negative value demonstrating the phylogenetically overdispersed [25]. Since all the phylogenetic diversity indices (i.e., Faith’s PD, SES.PD, MPD, and SES.PD) were not normally distributed based on Shapiro–Wilk tests, we used Wilcoxon rank-sum tests to identify their potential differences between four seasons (i.e., four months), as well as the potential differences between elevational sites (i.e., lowland and highland).

We performed Spearman’s rank correlations to test the correlations of pairwise microhabitat variables, and only one variable was considered if two or more variables exhibited a strong correlation (|r| > 0.70) [26]. Based on the results, ten variables were kept for further analyses, including elevation, air temperature, air humidity, water temperature, water pH, water depth, number of trees, leaf litter cover, rock cover, and current velocity. Generalized linear models (GLMs) were then used to explore the determination of variables to different phylogenetic diversity indices, separately. In the models, the phylogenetic diversity indices were considered as the dependent variables, and the ten selected environmental variables were the independent variables. The best-fitted model was selected according to the minimum corrected AIC values (AICc) because of the small sample size [27]. Finally, we also conducted hierarchical partitioning analyses to reveal the relative contribution of different variables in the best-fitted model to the variation of each phylogenetic diversity index.

All statistical analyses were conducted in R 3.6.1 [28]. The calculation of phylogenetic diversity indices was based on the *picante* package [29]. Shapiro–Wilk test was performed using the *stats* package [28]. Spearman’s rank correlation was performed using the *psych* package [30]. Wilcoxon test was performed using the *PMCMR* package [31]. GLMs were conducted using the *MuMIn* package [32]. Hierarchical partitioning analyses were undertaken using the *hier.part* package [33].

## 3. Results

A total of 25 species belonging to 8 families were detected during the whole year’s field work (Appendix A). The dominant species in the lowland area were *Odorrana schmackeri* and *Fejervarya multistriata*, accounting for 36.08% and 11.62% of the total number of individuals, respectively. The dominant species in the highland area were *Odorrana margaretae* (20.56%), *Leptobrachella oshanensis* (15.82%), and *Quasipaa boulengeri* (14.64%; Figure 1A). In April, the dominant species were *L. oshanensis* and *Leptobrachium boringii*, accounting for 48.87% and 17.45% of the total number of individuals, separately. *O. schmackeri* (27.61%) and *F. multistriata* (12.15%) were more abundant in June, while *O. schmackeri* (22.34%), *O. margaretae* (19.64%), and *Pseudohynobius flavomaculatus* (14.95%) were more abundant in August. Finally, *Q. boulengeri* (44.93%) and *Amolops sinensis* (30.43%) were the dominant species in October (Figure 1B). Some species were only detected in specific seasons, such as *Hyla gongshanensis wulingensis*, *Rana jiemuxiensis*, *Zhangixalus nigropunctatus*, and *Zhangixalus omeimontis* in April, and *Zhangixalus dennysi* and *Megophrys tuberogranulata* in August. In contrast, some species can be observed during the four sampling events, such as *A. sinensis*, *Bufo gargarizans*, *O. margaretae*, *P. flavomaculatus*, and *Q. boulengeri*.

For the elevational spatial patterns, all the amphibian phylogenetic diversity indices were overall lower in lowland transects than those in highland transects. However, these relationships were not significant (Figure 2). In terms of the seasonal temporal patterns, the highest value of Faith’s PD was observed in June, which was significantly higher than that in April. When controlling the effects of species richness, amphibian assemblages in April exhibited the highest SES.PD value, which was significantly higher than that in August. For MPD and SES.MPD, values in June and August were lower but not more significant than those in April and October. Interestingly, SES.MPD values were < 0 in June and August, but were > 0 in April and October (Figure 3).

In the best-fitted models, Faith’s PD was significantly and positively determined by rock cover. In addition, both SES.PD and MPD were significantly and positively correlated with rock cover, but significantly and negatively correlated with water temperature and water depth (Table 1). Based on the hierarchical partitioning analyses, Faith’s PD was best explained by rock cover (55.64%), the number of trees (20.63%), and elevation (17.07%). SES.PD was mainly explained by water temperature (54.02%), followed by rock cover (17.77%) and water depth (15.25%). Finally, water temperature (56.15%) was the most important contributor to the variation of MPD, followed by rock cover (20.23%) and water depth (15.72%; Figure 4).

## 4. Discussion

### 4.1. Elevational Spatial Difference in Amphibian Phylogenetic Structures

Our results indicated that amphibians exhibited distinct assemblages between low- and high-elevational sites. It is widely recognized that amphibian species have their own distribution ranges (i.e., elevational spatial niche), which are regulated by their thermal tolerance ranges [34]. Therefore, we argue that elevational generalists such as *B. gargarizans* and *Q. boulengeri* could have wider thermal tolerance ranges. In contrast, species that can only be observed either in lowland areas (e.g., *Amolops chunganensis* and *O. schmackeri*) or highland areas (e.g., *Pseudorana sangzhiensis* and *Megophrys sangzhiensis*) should be considered as elevational specialists, which had narrower thermal tolerance ranges [16,19,34]. However, the differences in amphibian phylogenetic diversity between low- and high-elevational sites were not significant, which was in contrast with previous studies showing that lowland areas usually contained higher phylogenetic diversity when compared with that in highland areas, e.g., [12,24]. This is probably because the elevation of the study area was not high enough to induce a strong environmental change that can promote the rapid evolution of amphibians. Interestingly, the mean SES.MPD value in lowland sites was <0 (i.e., phylogenetically overdispersed), indicating that the main mechanism underlying amphibian community assembly processes was limiting similarity [25]. This could be attributed to the diverse habitat types in the lowland areas, providing different shelters for various species. In contrast, the mean SES.MPD value was >0 (i.e., phylogenetically clustered) in highland areas, suggesting that environmental filtering was more important in shaping amphibian assemblages [25]. This is because the transects in highland sites were all small forest streams, allowing phylogenetically similar species to live.

### 4.2. Seasonal Temporal Difference in Amphibian Phylogenetic Structures

Amphibian phylogenetic diversity also varied between four seasons, as amphibian assemblages were different in April, June, August, and October. Indeed, amphibian species occupied their own seasonal temporal niche, in accordance with their unique activities such as breeding, foraging, and migration [35]. For instance, the breeding season of *L. boringii* in this area is April [36,37], when many individuals can be observed in the transects. *Megophrys sangzhiensis* and *Pseudorana sangzhiensis* can only be detected in June and August, as they need to migrate into the habitat to forage and breed during this period [36,37]. Specifically, the highest Faith’s PD value was observed in June, which could be attributed to the highest species richness detected in June. Interestingly, the highest SES.PD value occurred in April, demonstrating that amphibian phylogenetic diversity was higher in this season when controlling the effects of species richness. These contrasting results between PD and SES.PD were consistent with previous studies, e.g., [12,24], suggesting that species richness should not be neglected when quantifying communities’ phylogenetic structures [5,13]. Moreover, the mean SES.MPD values in June and August were <0, showing that limiting similarity drove the assembly of amphibian assemblages. However, the mean SES.MPD values in April and October were >0, demonstrating that the main mechanism was environmental filtering. These results, combined with the unique amphibian assemblages in each season, suggest that seasonal fluctuation should be considered in community assembly studies.

### 4.3. Microhabitat Determinants of Amphibian Phylogenetic Structures

In the present study, we found that streams with high rock cover, low water temperature, and shallow water bodies contained high phylogenetic diversity. Most of these streams were distributed in the high elevations, associated with high species richness of amphibians in the study area [16]. Since Faith’s PD values are positively correlated with species richness [9], it is not surprising that Faith’s PD is significantly and positively explained by rock cover. Both SES.PD and MPD values were significantly and negatively determined by water temperature and water depth. This is because these environmental conditions can harbor some distantly related species such as *L. boringii* and *O. margaretae*, which preferred relatively low water temperatures and shallow water bodies [36,37].

## 5. Conclusions

Overall, the present study investigated the elevational spatial and seasonal temporal patterns of amphibian phylogenetic structures in temperate montane streams. Our results revealed various amphibian phylogenetic patterns between lowland and highland sites, as well as between four seasons, associated with distinct community assembly rules (phylogenetically overdispersed vs. phylogenetically clustered). These patterns should be determined by the different amphibian assemblages detected in the streams between elevational sites and seasons. More importantly, these patterns were also strongly affected by microhabitat variables such as rock cover, water temperature, and water depth. Our results can provide fundamental knowledge to better protect amphibian diversity. Both elevational and seasonal variations are important to understanding the general patterns of amphibian community assembly rules. Since this study was conducted within a limited number of transects, more streams should be incorporated in future studies. Moreover, since this study can only reflect random effects of the sampling method plus seasonal variation of microhabitat use, long-term monitoring is needed. In addition, amphibian functional structures need to be incorporated in future studies to better understand the community assembly mechanisms in the ecosystems.

## Figures and Tables

**Figure 1 animals-12-01673-f001:**
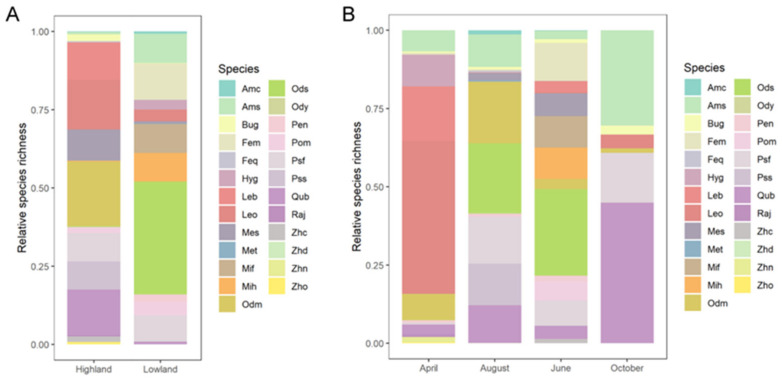
Elevational spatial and seasonal temporal compositions of amphibian species. (**A**) Amphibian composition in highland and lowland sites. (**B**) Amphibian composition in four different months. Species abbreviations are as follows: Amc: *Amolops chunganensis*, Ams: *Amolops sinensis*, Bug: *Bufo gargarizans*, Fem: *Fejervarya multistriata*, Feq: *Feirana quadranus*, Hyg: *Hyla gongshanensis wulingensis*, Leo: *Leptobrachella oshanensis*, Leb: *Leptobrachium boringii*, Mes: *Megophrys sangzhiensis*, Met: *Megophrys tuberogranulata*, Mif: *Microhyla fissipes*, Mih: *Microhyla heymonsi*, Odm: *Odorrana margaretae*, Ods: *Odorrana schmackeri*, Ody: *Odorrana yizhangensis*, Pen: *Pelophylax nigromaculatus*, Pom: *Polypedates megacephalus*, Psf: *Pseudohynobius flavomaculatus*, Pss: *Pseudorana sangzhiensis*, Qub: *Quasipaa boulengeri*, Raj: *Rana jiemuxiensis*, Zhc: *Zhangixalus chenfui*, Zhd: *Zhangixalus dennysi*, Zhn: *Zhangixalus nigropunctatus*, Zho: *Zhangixalus omeimontis*.

**Figure 2 animals-12-01673-f002:**
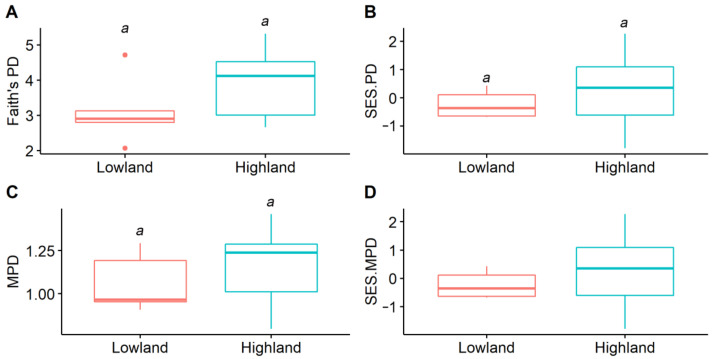
Amphibian phylogenetic structures between lowland and highland streams. (**A**) Faith’s PD, (**B**) SES.PD (standardized effect size of Faith’s PD), (**C**) MPD (mean pairwise phylogenetic distance index), (**D**) SES.MPD (standardized effect size of MPD). The same letter on top of the error bars indicate no significant difference between lowland and highland sites.

**Figure 3 animals-12-01673-f003:**
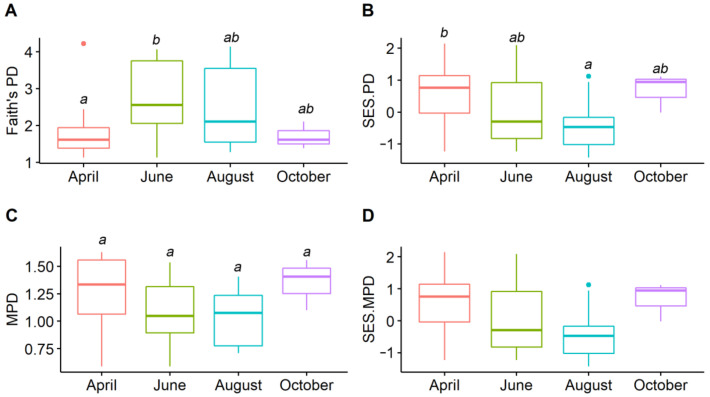
Amphibian phylogenetic structures between four months in montane streams. (**A**) Faith’s PD, (**B**) SES.PD (standardized effect size of Faith’s PD), (**C**) MPD (mean pairwise phylogenetic distance index), (**D**) SES.MPD (standardized effect size of MPD). Different letters on top of the error bars indicate a significant difference between pairwise seasons.

**Figure 4 animals-12-01673-f004:**
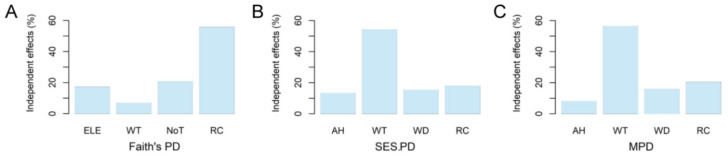
The independent contributions of selected environmental variables to the variation of different phylogenetic structure indices based on hierarchical partitioning analyses. (**A**) Faith’s PD; (**B**) SES.PD; (**C**) MPD. Details of the abbreviations are as follows: SES.PD, standardized effect size of Faith’s PD; MPD, mean pairwise phylogenetic distance index; SES.MPD, standardized effect size of MPD; ELE, elevation; WT, water temperature; NoT, the number of trees; RC, rock cover; AH, air humidity; WD, water depth.

**Table 1 animals-12-01673-t001:** The best-fitted models selected by the generalized linear models (GLMs) for phylogenetic structure indices. Significant *p* values are in bold. Details of the abbreviations are as follows: ELE: elevation, AT: air temperature, AH: air humidity, WT: water temperature, pH: water pH, WD: water depth, NoT: the number of trees, LLC: leaf litter cover, RC: rock cover, CV: current velocity.

Indices		Intercept	ELE	AT	AH	WT	pH	WD	NoT	LLC	RC	CV
Faith’s PD	Estimate	−0.029	<0.001	/	/	0.065	/	/	0.003	/	0.019	/
	Std.Error	1.037	<0.001	/	/	0.046	/	/	0.002	/	0.005	/
	*p*	0.978	0.125	/	/	0.165	/	/	0.155	/	**<0.001**	/
SES.PD	Estimate	4.293	/	/	−0.019	−0.148	/	−0.009	/	/	0.007	/
	Std.Error	0.980	/	/	0.010	0.031	/	0.003	/	/	0.004	/
	*p*	<0.001	/	/	0.052	**<0.001**	/	**0.007**	/	/	0.139	/
MPD	Estimate	2.182	/	/	−0.003	−0.043	/	−0.003	/	/	0.002	/
	Std.Error	0.277	/	/	0.003	0.009	/	<0.001	/	/	0.002	/
	*p*	<0.001	/	/	0.178	**<0.001**	/	**0.005**	/	/	0.071	/

## Data Availability

The datasets presented in this study are available from the corresponding author on reasonable request.

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
