# Peer review of "Distinct Amphibian Elevational and Seasonal Phylogenetic Structures Are Determined by Microhabitat Variables in Temperate Montane Streams"

_animals, 2022, doi:10.3390/ani12131673_

Round 1
Reviewer 1 Report
Language must be improved throughout the manuscript, including the Abstract, which contains some awkward sentences, as well as repetitions such as using “determined” in two subsequent sentences.
Simple Summary and Abstract are too similar. The simple summary should be written in a simpler way as there is too much redundancy otherwise. Furthermore, the abstract should also include information where Tianping mountain is located.
Abstract: The conclusion “Therefore, habitat environment should be well protected to maintain high phylogenetic diversity.” is too general and, therefore, too weak. More specific measures should be suggested.
Abbreviations like PD, SES etc. must be explained upon first mentioning in the abstract as well as in the manuscript body.
Introduction: “Traditionally, biodiversity was assessed based on its taxonomic facet, which considered species to be ecological equivalence.” is a strange statement and the second part does not make sense. There are many good papers defining the different levels of biodiversity and taxonomy, such as Lücking R (2020) Three challenges to contemporaneous taxonomy from a licheno-mycological perspective. Megataxa, 1(1), 78–103. doi: 10.11646/megataxa.1.1.16
2.3 Microhabitat variables: units should be given for all variables measured.
2.4 Statistical analyses: the first part of this section describes the origin of the data used for constructing the phylogenetic tree. This is not part of the statistical analyses. Furthermore, data from public databases such as GenBank should generally be used with extreme caution. A published example illustrating the risks of using data from GenBank is the paper “Shark tales: A molecular species-level phylogeny of sharks”, published 2011 in “Molecular Phylogenetics and Evolution”, with very obvious misidentifications and misinterpretations. If using data from NCBI, the authors should critically discuss the risks of using data without verification of the identification and include this in the discussion of the limitations of their study. Furthermore, it must be clearly indicated which data are from the present study and which are from NCBI.
2.4 Statistical analyses: The equation (not formula) given must be written using minus signs instead of n-dashes.
3. Results: It seems like all 25 species included in the phylogenetic analyses were caught and sequenced in the present study. If this is correct, why were sequences from NCBI used at all? It seems like the authors have enough own data. Please explain.
Captions to Figure 2 and 4: Abbreviations used in these figures must be explained in the captions so that they can be understood without having read the manuscript text.
4. Discussion: A discussion of the limitations of the present study is missing (see also first comment on section 2.4).
5. Conclusions, line 310: delete “the”. Furthermore, the conclusions need to be rephrased to become more concise and meaningful.
Data availability: The sequence data obtained in the present study should be uploaded in GenBank.
References: Although references are numbered consecutively throughout the manuscript, the list of references appears to be alphabetical. This appears to be erroneous, please check.
Author Response
Reviewer 1:
Language must be improved throughout the manuscript, including the Abstract, which contains some awkward sentences, as well as repetitions such as using “determined” in two subsequent sentences.
Reply: The language has been carefully revised throughout the whole manuscript, in particular the Abstract. And we also reduced the utilization of some words.
Simple Summary and Abstract are too similar. The simple summary should be written in a simpler way as there is too much redundancy otherwise. Furthermore, the abstract should also include information where Tianping mountain is located.
Reply: The Simple Summary section has been rewritten, and the location of Tianping mountain has been provided in the Abstract in the revised manuscript (L. 33).
Abstract: The conclusion “Therefore, habitat environment should be well protected to maintain high phylogenetic diversity.” is too general and, therefore, too weak. More specific measures should be suggested.
Reply: This sentence has been removed, and new conclusions have been provided in the revised manuscript (L. 44-46).
Abbreviations like PD, SES etc. must be explained upon first mentioning in the abstract as well as in the manuscript body.
Reply: Done (L. 37-38).
Introduction: “Traditionally, biodiversity was assessed based on its taxonomic facet, which considered species to be ecological equivalence.” is a strange statement and the second part does not make sense. There are many good papers defining the different levels of biodiversity and taxonomy, such as Lücking R (2020) Three challenges to contemporaneous taxonomy from a licheno-mycological perspective. Megataxa, 1(1), 78–103. doi: 10.11646/megataxa.1.1.16
Reply: This sentence has been revised, and the suggested reference has been cited (L. 57-58).
2.3 Microhabitat variables: units should be given for all variables measured.
Reply: Done (L. 144-150).
2.4 Statistical analyses: the first part of this section describes the origin of the data used for constructing the phylogenetic tree. This is not part of the statistical analyses. Furthermore, data from public databases such as GenBank should generally be used with extreme caution. A published example illustrating the risks of using data from GenBank is the paper “Shark tales: A molecular species-level phylogeny of sharks”, published 2011 in “Molecular Phylogenetics and Evolution”, with very obvious misidentifications and misinterpretations. If using data from NCBI, the authors should critically discuss the risks of using data without verification of the identification and include this in the discussion of the limitations of their study. Furthermore, it must be clearly indicated which data are from the present study and which are from NCBI.
Reply: The description of the phylogenetic tree construction has been listed as a new subsection namely “Phylogenetic tree” (L. 154-161). In addition, we apologize as when we checked the data used, we found that all the sequences we used were from our own sample. This has been clearly indicated in the revised manuscript (L. 157-159).
2.4 Statistical analyses: The equation (not formula) given must be written using minus signs instead of n-dashes.
Reply: Actually they are the minus signs.
- Results: It seems like all 25 species included in the phylogenetic analyses were caught and sequenced in the present study. If this is correct, why were sequences from NCBI used at all? It seems like the authors have enough own data. Please explain.
Reply: We apologize for this mistake. Actually all the data used in the present study was indeed from our own samples. This has been clearly indicated in the revised manuscript (L. 157-159).
Captions to Figure 2 and 4: Abbreviations used in these figures must be explained in the captions so that they can be understood without having read the manuscript text.
Reply: Done (L. 260-274).
- Discussion: A discussion of the limitations of the present study is missing (see also first comment on section 2.4).
Reply: We have added some limitations of this study in the revised manuscript (L. 345-346).
- Conclusions, line 310: delete “the”. Furthermore, the conclusions need to be rephrased to become more concise and meaningful.
Reply: Done (L. 336; L. 342-349).
Data availability: The sequence data obtained in the present study should be uploaded in GenBank.
Reply: We have uploaded the sequences in GenBank. However, they are still in processing (see the figure below) . We will provided the link into the manuscript once we receive the feedback.
References: Although references are numbered consecutively throughout the manuscript, the list of references appears to be alphabetical. This appears to be erroneous, please check.
Reply: We apologize for this mistake. The references have been carefully checked.

Reviewer 2 Report
The study aims to unravel the spatial and temporal phylogenetic diversity in two sites in a montane forest. While the spatial diversity usually has a phylogenetic component, it does not seem to have one in this study area (this not clear from the text, presentation must be clearer). The "seasonal" phylogenetic diversity detected is nonsense to my opinion because it is trivial that you won´t be able to detect all species inhabiting an area in any season. But still, all species are present in the area (not within the transect), so phylogenetic diversity cannot have a seasonal structure. If you compare decades or longer periods, you may reveal such a structure, but surely not within a year. The conclusion that certain types of microhabitat correlate with diversity are more than trivial and well known. In conclusion, I believe that this ms needs a thorough rewriting and reflection upon the focus.
Minor issues: stated in the attached ms-pdf. Please check style of reference list, in the text it is by numbers, but number in the list of references obviously do not correspond to those in the text.

Author Response
Reviewer 2:
The study aims to unravel the spatial and temporal phylogenetic diversity in two sites in a montane forest. While the spatial diversity usually has a phylogenetic component, it does not seem to have one in this study area (this not clear from the text, presentation must be clearer). The "seasonal" phylogenetic diversity detected is nonsense to my opinion because it is trivial that you won´t be able to detect all species inhabiting an area in any season. But still, all species are present in the area (not within the transect), so phylogenetic diversity cannot have a seasonal structure. If you compare decades or longer periods, you may reveal such a structure, but surely not within a year. The conclusion that certain types of microhabitat correlate with diversity are more than trivial and well known. In conclusion, I believe that this ms needs a thorough rewriting and reflection upon the focus.
Reply: What we really want to test is the amphibian phylogenetic diversity difference between low and high elevational montane streams, as well as the amphibian phylogenetic diversity difference between four seasons in the montane streams. We think this work can provide some important information to understand the assembly of amphibians. Actually, many previous studies have quantified the seasonal change of amphibian/reptile diversity (e.g., species richness, functional diversity, and phylogenetic diversity) in the transects (Strauß et al. 2016, Caldas et al. 2019, Zhu et al. 2020, Sun et al. 2021, Barros et al. 2022). We have revised the title of the present study to make it more accurate.
Minor issues: stated in the attached ms-pdf. Please check style of reference list, in the text it is by numbers, but number in the list of references obviously do not correspond to those in the text.
Reply: We apologize for this mistake. The references have been carefully checked.
Simple Summary
Is this section really needed?
Reply: This section is required by the journal, which has been rewritten in the revised manuscript (L. 13-28).
- 35-36: Do they really? In the paper you state that differences are not significantly different.
Reply: This part has been revised to make it more accurate (L. 39).
- 37: which ones?
Reply: More details have been provided (L. 41).
- 49: I believe that the list of references (alphabetical order) does not correspond to the number system given in the text. Please correct or adopt alphabetical system in the text.
Reply: We apologize for this mistake. The references have been carefully checked.
- 70: “can be” should be “were”
Reply: Done (L. 78).
- 82: “formulation” could be “detection”
Reply: Done (L. 91).
- 134: be more precise. In the shade? at 2m above ground
Reply: Air temperature was measured at 2m above the ground. This has been provided in the revised manuscript (L. 144-145).
- 170-172: consider to use a multivariate factorial analyses to look for significant environmental factors, unrelated to each other. Then, use the factors instead of the empirically measured variables.
Reply: If we were right, the Reviewer 2 suggested to use analyses like PCA? We totally agree that both GLMs and PCA analyses can be used to explore the relationships between environmental variables and diversity indices. However, compared to PCA, GLMs could explain a higher percentage of total variation and is a quantitative prediction (Zhu & Kang, 2005), which is widely used in previous related studies (e.g., Borthagaray et al., 2020; Carvajal-Quintero et al., 2015; He et al., 2020). Therefore, we suggested to keep using GLMs.
- 193: remove “and”
Reply: Done (L. 212).
- 200: remove “interestingly”
Reply: Done (L. 219).
- 224: remove “3.1 Figures and Tables”
Reply: Done (L. 243).
- 252-254: trivial
Reply: This sentence has been revised (L. 277-279).
- 262: contradiction to the first sentence of discussion
Reply: True. Therefore, the first sentence of the discussion has been revised (L. 277-279).
- 276: This is simply an artifact of the sampling method. All species detected at the transect are present in the area during the whole year, but not always at the transect. So do not overinterprete the season phylogenetic diversity.
Reply: The season fluctuation of amphibian assemblages in the transects can reflect the distinct ecological niche occupied by different species. And the seasonal phylogenetic diversity in the transects can provide some important information to understand the assembly of amphibians. Please also see the reply before.
- 284: I don´t understand the meaning of this statement
Reply: This sentence has been removed (L. 309).
- 311: state explicitly the suppossed assembly rules.
Reply: Done (L. 338).
- 312-313: meaning enigmatic
Reply: This sentence has been revised to make it more clear (L. 338-340).
- 315-316: again, trivial. No "phylogenetic" study is needed to underpin this conclusion.
Reply: This part has been revised to make it more accurate (L. 342-347).
- 389: da Silva
Reply: Done (L. 415).
Round 2
Reviewer 1 Report
Dear authors, thank you very much for carefully revising the manuscript. Although it has been improved significantly, the language still needs to be revised throughout, particularly concerning the revised phrases (e.g. "phylogenetically overdispersion" is not correct and in line 65/66 "amphibians" need an apostrophe, i.e. "amphibians'"). And there are more mistakes throughout the manuscript.
Author Response
Reviewer 1:
Dear authors, thank you very much for carefully revising the manuscript. Although it has been improved significantly, the language still needs to be revised throughout, particularly concerning the revised phrases (e.g. "phylogenetically overdispersion" is not correct and in line 65/66 "amphibians" need an apostrophe, i.e. "amphibians'"). And there are more mistakes throughout the manuscript.
Reply: We appreciate that Reviewer 1 agree with our revisions. Actually, “phylogenetically overdispersion” is the word that has been widely used in many previous phylogenetic structure studies (e.g., Yang et al. 2014, Ding et al. 2021, Wang et al. 2022). In addition, we have also asked one of our colleagues to carefully revised the language throughout the whole manuscript.
Reviewer 2 Report
The ms has been revised according to the comments made on the earlier version. However, I still question the sense of the "seasonal phylogenetic structure" and insist that it is merely the reflection of random effects of the sampling method plus seasonal variation of microhabitat use. Therefore, I recommend strongly to add at least a short reflection on this issue in the discussion.
Author Response
Reviewer 2:
The ms has been revised according to the comments made on the earlier version. However, I still question the sense of the "seasonal phylogenetic structure" and insist that it is merely the reflection of random effects of the sampling method plus seasonal variation of microhabitat use. Therefore, I recommend strongly to add at least a short reflection on this issue in the discussion.
Reply: We appreciate that Reviewer 2 agree with our revisions. Following Reviewer 2’s suggestions, we have added some discussion related to the limitation of this study in the revised manuscript (L. 318-320).
References:
Ding, Z., H. Hu, M. W. Cadotte, J. Liang, Y. Hu, and X. Si. 2021. Elevational patterns of bird functional and phylogenetic structure in the central Himalaya. Ecography:ecog.05660.
Wang, X.-Y., M.-J. Zhong, J. Zhang, X.-F. Si, S.-N. Yang, J.-P. Jiang, J.-H. Hu. Multidimensional amphibian diversity and community structure along a 2 600 m elevational gradient on the eastern margin of the Qinghai-Tibetan Plateau. Zoological Research 43:40–51.
Yang, J., G. Zhang, X. Ci, N. G. Swenson, M. Cao, L. Sha, J. Li, C. C. Baskin, J. W. F. Slik, and L. Lin. 2014. Functional and phylogenetic assembly in a Chinese tropical tree community across size classes, spatial scales and habitats. Functional Ecology 28:520–529.